# The impact of COVID-19 on cancer screening and treatment in older adults: The Multiethnic Cohort Study

**Victoria P Mak[1]\*, Kami White[1], Lynne R Wilkens[1], Iona Cheng[2], Christopher A Haiman[3], Loic Le Marchand[1]**

[1]Population Sciences in the Pacific Program (Cancer Epidemiology), University of Hawaii Cancer Center, University of Hawaii at Manoa, Honolulu, United States; [2]Department of Epidemiology and Biostatistics, University of California, San Francisco, San Francisco, United States; [3]Center for Genetic Epidemiology, University of Southern California, Los Angeles, United States

## ABSTRACT

**Background:** The Coronavirus Disease of 2019 (COVID-19) has impacted the health and day-to-day life of individuals, especially the elderly and people with certain pre-existing medical conditions, including cancer. The purpose of this study was to investigate how COVID-19 impacted access to cancer screenings and treatment, by studying the participants in the Multiethnic Cohort (MEC) study.

**Methods:** The MEC has been following over 215,000 residents of Hawai'i and Los Angeles for the development of cancer and other chronic diseases since 1993–1996. It includes men and women of five racial and ethnic groups: African American, Japanese American, Latino, Native Hawaiian, and White. In 2020, surviving participants were sent an invitation to complete an online survey on the impact of COVID-19 on their daily life activities, including adherence to cancer screening and treatment. Approximately 7,000 MEC participants responded. A cross-sectional analysis was performed to investigate the relationships between the postponement of regular health care visits and cancer screening procedures or treatment with race and ethnicity, age, education, and comorbidity.

**Results:** Women with more education, women with lung disease, COPD, or asthma, and women and men diagnosed with cancer in the past 5 years were more likely to postpone any cancer screening test/procedure due to the COVID-19 pandemic. Groups less likely to postpone cancer screening included older women compared to younger women and Japanese American men and women compared to White men and women.

**Conclusions:** This study revealed specific associations of race/ethnicity, age, education level, and comorbidities with the cancer-related screening and healthcare of MEC participants during the COVID-19 pandemic. Increased monitoring of patients in high-risk groups for cancer and other diseases is of the utmost importance as the chance of undiagnosed cases or poor prognosis is increased as a result of delayed screening and treatment.

**Funding:** This research was partially supported by the Omidyar 'Ohana Foundation and grant U01 CA164973 from the National Cancer Institute.

\*For correspondence: vmak@hawaii.edu

**Competing interest:** The authors declare that no competing interests exist.

## Editor's evaluation

The authors used the Multiethnic Cohort (MEC) study to study how COVID-19 impacted access to cancer screenings and treatment. This study's important findings served to identify key factors associated with cancer-related screening and healthcare-seeking during the pandemic. This investigation provides solid evidence to inform future policies, particularly in older and vulnerable populations.

## Introduction

The highly contagious Coronavirus Disease of 2019 (COVID-19) has impacted the health and day-to-day life of individuals, especially the elderly and people with certain preexisting medical conditions, including cancer. COVID-19 was declared a worldwide pandemic on March 11, 2020, after its discovery in late December 2019. As of March 10, 2023, 6,881,955 deaths had been reported among 676,609,955 cases worldwide (*Dong et al., 2020*). Due to the rapid spread and high infectivity rates, stay-at-home orders were decreed to allow for physical distancing and self-isolation. Populations at high risk of contracting COVID-19 and developing severe illness were the elderly and people of all ages with pre-existing medical conditions (such as diabetes, high blood pressure, heart disease, lung disease, or cancer) (*WHO, 2020*).

Cancer patients were not only at a higher risk of severe illness from COVID-19 but changes in cancer care were also reported. One example is the finding of the COVID-19 and Cancer Outcomes Study, which is a multicenter prospective cohort study comprised of adult patients with a current or history of hematological malignancy or invasive solid tumor, who were scheduled for an outpatient medical oncology visit at the Tisch Cancer Institute in New York City or the Dana-Farber Cancer Institute in Boston. Researchers found that 10% of patients in the cohort had treatment delayed during the pandemic and, of that subset, 48% had treatment delayed due to a COVID-19 diagnosis (*Schmidt et al., 2020*). In the United States, COVID-19 disproportionately affected racial and ethnic minorities, particularly African Americans, with an observed two-fold higher rate of hospitalization and a greater than two-fold higher rate of death, as compared to Non-Hispanic Whites (*Newman et al., 2021*). The disparity seen with COVID-19 was consistent with patterns of disparities observed for cancer; it is well-documented that 5-year survival rates for multiple cancers are lower in African Americans compared to White Americans. While cancer is a multifactorial disease that is influenced by genetic and environmental factors, COVID-19 is an infectious disease that is enabled by the cellular expression of angiotensin-converting enzyme 2 (ACE2) receptors. However, preexisting comorbidities, socioeconomic disadvantages, living conditions, health literacy, and access to health care appear to fuel underlying risks for both cancer and COVID-19 disparities (*Newman et al., 2021*).

In addition, the pandemic has affected access to cancer screenings, especially for colorectal cancer (CRC) and breast cancer. The recommendation for CRC screening among average-risk adults aged 50–75 years is to undergo an annual fecal immunochemical test (FIT) or colonoscopy every 10 years (*Corley et al., 2021*). Screening and appropriate follow-up can reduce incidence and mortality, which is important because CRC is the second leading cause of cancer death in the United States (*Nodora et al., 2021*). One study found that CRC screening rates are particularly low in Hispanic and Asian American adults, recent immigrants, those with low incomes, and the uninsured (*Nodora et al., 2021*). During the COVID-19 pandemic, colorectal cancer screenings were reported to have dropped by 84.5% at a network of 20 United States institutions with more than 28 million patients (*London et al., 2020*).

For breast cancer, which is the most common non-cutaneous malignancy and the second most lethal tumor among women in the US, the recommendation is biennial mammography in women ages 50–74 years (*Corley et al., 2021*). In the MEC, Native Hawaiian participants have the highest risk of breast cancer, followed by Japanese American and White participants (*Pike et al., 2002*). Nationwide, the breast cancer mortality rate is 40% higher for African American patients compared with White patients (*Siegel et al., 2020*). During the COVID-19 pandemic, breast cancer screenings dropped by 89.2% at a network of 20 United States institutions with more than 28 million patients (*London et al., 2020*).

Lastly, studies have suggested that educational attainment is associated with adherence to cancer screening. For breast cancer, a positive association was found between education level and adherence to mammography screening (*Damiani et al., 2015*). For colorectal cancer, a study on United States veterans aged 50–75 years concluded that both education and income show a statistically significant dose-dependent direct association with screening (*Rodriguez and Smith, 2016*).

In summary, COVID-19 has impacted medical care delivery, access to cancer screening, and the day-to-day life of individuals worldwide. However, the full impact of the pandemic on cancer screenings and cancer care is not entirely known. The purpose of this cross-sectional study was to understand how COVID-19 affected access to cancer screening and treatment, by studying the participants in the Multiethnic Cohort (MEC) study.

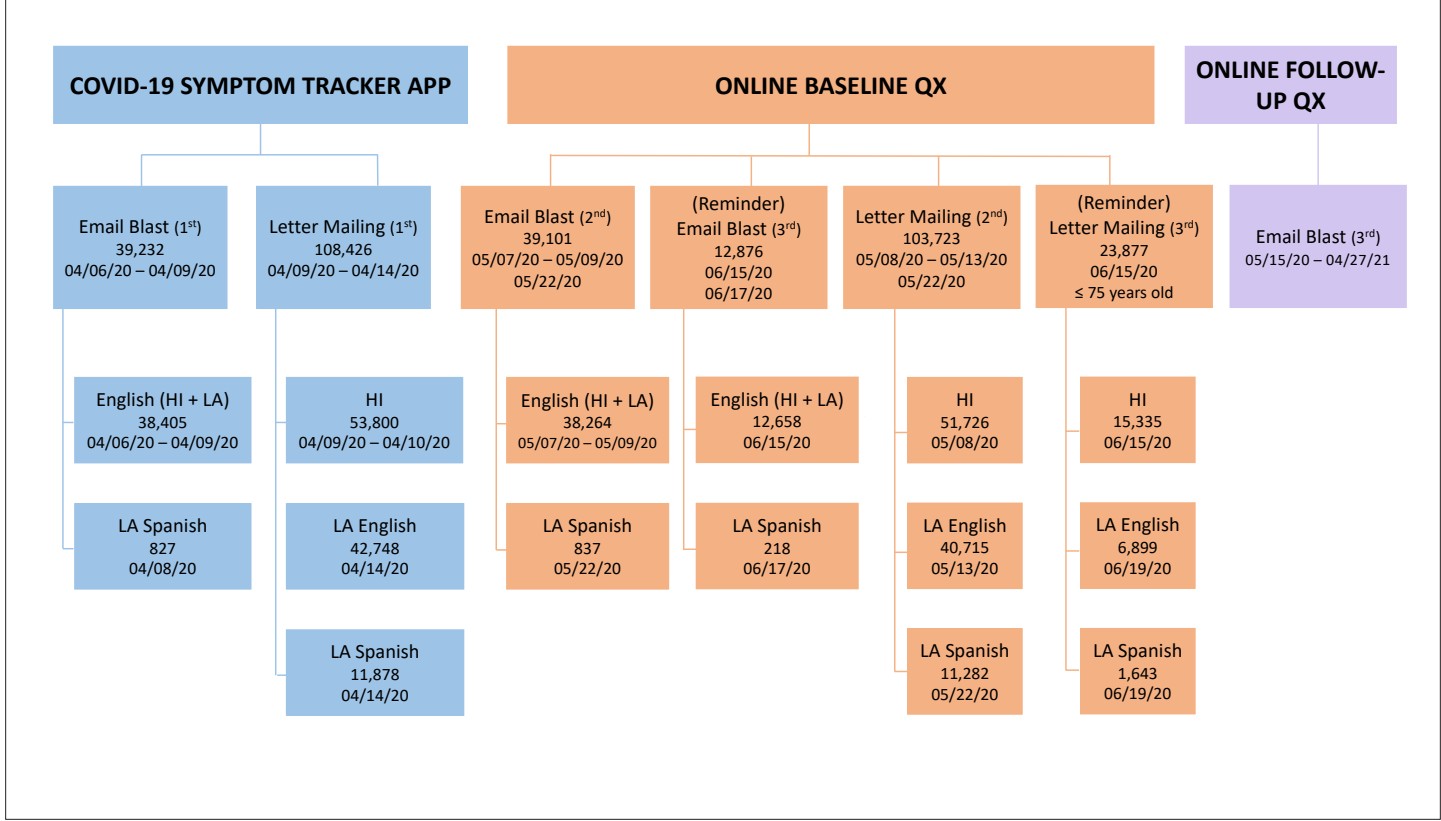

**Figure 1.** Timeline of the MEC COVID-19 Study.

## Methods

### Study population

The Multiethnic Cohort (MEC) was established in 1993–1996 at the University of Hawai'i Cancer Center (UHCC) and the University of Southern California (USC) with the goals of elucidating lifestyle and genetic risk factors responsible for explaining the ethnic/racial disparities that exist for cancer and other chronic diseases (*Kolonel et al., 2000*). The cohort has followed over 215,000 residents of Hawai'i and Los Angeles, aged 45–75 years old at recruitment, and includes men and women of five main ethnic and racial origins: African American, Japanese American, Latino, Native Hawaiian, and White. Participants entered the cohort by returning a 26-page mailed questionnaire. They also were sent follow-up questionnaires every five years. Disease ascertainment was primarily conducted by linking the cohort to SEER cancer registries, death certificate registries, CA hospital discharge diagnoses, and Medicare. Approximately 100,000 MEC members (median age: 82) were still alive as of 2019. Demographics (i.e. race/ethnicity, maximum years of education attained, birthplace), pre-existing and incident disease outcomes, and lifestyle risk factors were available from the MEC database.

### Study design

In May 2020, MEC participants were invited to participate in an online survey (implemented in Qualtrics) on the impact of COVID-19 on their everyday life and health-related behaviors. Emails were sent to over 39,000 and a letter to ~100,000 MEC participants. The initial and a second invitation were sent to all living MEC participants, while a third invitation was sent to those who were 75 or younger. A paper survey was provided if requested. Participants registered online by providing their Email Address, First name, Middle name, Last name, Sex, Birth year, and Zip Code. The entered data linked each participant back to the MEC database. A Spanish registration page was available for registration in Spanish. Each participant was sent a unique identifier and login to answer the survey online after providing informed consent. Participants who answered the baseline survey were invited to fill out a

weekly, then, monthly shorter follow-up online survey (*Figure 1*). Paper follow-up surveys were not available. The surveys started in May 2020 and ended in April 2021. The last baseline survey (the last opportunity to enter the study) was collected on September 25, 2020. The survey was approved by the University of Hawaii (CHS 9575) and University of Southern California (HS-17–00714) Institutional Review Boards.

## Statistical analysis

All baseline surveys with at least 50% completion were used for this analysis. Descriptive statistics were utilized to summarize patient demographics, including means ± standard deviations (SD) for continuous variables and frequencies and percentages for categorical variables. A cross-sectional analysis was performed to investigate the relationships between the outcomes of postponement of regular health care visits and cancer screening tests/procedures/treatment with race and ethnicity, age, education, and comorbidity. Chi-square tests and binary logistic regression models of the outcomes of postponement and screening were used to calculate odds ratios (and 95% confidence intervals) and/or p-values (≤0.05 was taken as statistically significant) in each sex to describe these relationships using the Statistical Analysis System (SAS), version 9.4. Whites were taken as the reference group when comparing associations across racial and ethnic groups. For comorbidities, the reference was reporting no comorbidity.

## Results

*Table 1* presents the number of COVID-19 survey participants by various characteristics. A total of 6,974 MEC members answered the baseline survey, 6,068 via the online survey, and 906 via the paper survey. Fewer men (43.5%) than women (56.5%) responded. 76.6% of COVID survey participants were from Hawai'i and 23.6% were from Los Angeles. The survey response rate was 7.2% for men and 6.3% for women. The racial/ethnic group with the highest response rates was White (13.2%) for men; 12% for women, followed by Japanese American (8.9; 7.4), Native Hawaiian (7.3; 7.5), Other (2.7; 4.5), African American (2.7; 2.8), and Latino (2.2; 2.0). In *Supplementary file 1*, the distribution of the MEC cohort survivors in 2019 is compared with the distribution of the COVID survey participants in 2020 by race and education. For both males and females, the COVID survey respondents included a larger representation of Whites and Japanese Americans and a smaller representation of Latinos, African Americans, and Others compared to the entire 2019 surviving cohort. The proportion of the sample that was Native Hawaiians was similar to the MEC survivors. For education, a greater percentage of COVID-19 survey respondents had at least some college education compared to the entire 2019 surviving cohort. This pattern of higher attained education was observed for each racial/ethnic group.

Among participants in the baseline COVID-19 survey, the most common race/ethnicity reported was White (45.2% of men; 42.7% of women), followed by Japanese American (34.9%; 30.3%), Latino (7.9%; 7.4%), Native Hawaiian (6.4%; 8.1%), African American (3.5%; 6.5%), and Other (2.0%; 4.9%). The ethnicity category of Other comprised a majority of Chinese Americans (38.1%) and Filipino (36.6%). The mean age and education were similar in both sexes; however, men were more often smokers and alcohol drinkers and had a higher BMI than women. Women had a greater mean Healthy Eating Index-2010 score than men as reported in the MEC baseline questionnaire (1993–1996).

Overall, 78.8% of men and 84.4% of women reported making changes to their lifestyle or daily activities due to COVID-19 since March 2020. Examples of activities were specifically about COVID-19 prevention measures such as covering a cough or sneeze, more frequent hand washing, and social distancing. Among men, 4.2% and 6.3% reported drinking more and drinking less during the COVID-19 pandemic, respectively. The corresponding %'s for women were 4.3 and 5.2. Only 3.7% of men and 2.6% of women were current smokers.

Regarding the participant's health status, 3.5% and 19.7% of men had health problems that require them to stay at home and limit their activities, respectively, at the time of the COVID-19 baseline survey (*Supplementary file 2*). The corresponding %'s for women were 4.9 and 19.3. In addition, the survey respondents had a wide range of comorbidities in 2020, with hypertension (48% of men; 47.1% of women) being the most prevalent, followed by diabetes (18.3%; 14.4%), and heart disease (20.8%; 10.6%). Lastly, the most frequently used medication was aspirin (37.9% of men; 28.1% of women),

Table 1. Demographics of baseline COVID survey participants from Hawai'i and Los Angeles (May-September 2020, N=6974).

| | | Male<br>N (%) or Mean ±SD | Female<br>N (%) or Mean ±SD |
|---|---|---|---|
| | | 3034 (43.5) | 3940 (56.5) |
| Race/Ethnicity | White | 1371 (45.2) | 1683 (42.7) |
| | Japanese American | 1059 (34.9) | 1195 (30.3) |
| | Latino | 241 (7.9) | 291 (7.4) |
| | Native Hawaiian | 194 (6.4) | 320 (8.1) |
| | African American | 107 (3.5) | 257 (6.5) |
| | Other* | 62 (2.0) | 194 (4.9) |
| Age (years) | 66–72.8 | 715 (23.57) | 946 (24.01) |
| | 72.8–74.6 | 787 (25.94) | 1008 (25.58) |
| | 74.6–77.9 | 796 (26.24) | 979 (24.85) |
| | 77.9–102 | 736 (24.26) | 1007 (25.56) |
| Age (years) | | 76±4.9 | 76±5.3 |
| Years of Education †,‡ | | 15.86±2.1 | 15.48±2.2 |
| Pack Years†,‡ (MEC baseline 1993–1996) | | 8.57±12.4 | 5.16±9.9 |
| Alcohol Intake†,‡ (g/day) (MEC baseline 1993–1996) | | 12.65±20.7 | 5.15±11.8 |
| Number of Days an Alcoholic Beverage was Consumed in the Past 2 Weeks†,‡ | | 4.61±5.6 | 2.83±4.7 |
| Number of Days with 5 or More Drinks Consumed on the Same Occasion in the Past 2 Weeks†,‡ | | 0.37±1.8 | 0.07±0.8 |
| Dietary Patterns Healthy Eating Index-2010 score†,‡ (MEC Baseline 1993–1996) | | 64.95±10.8 | 68.66±10.5 |
| Made Lifestyle and Healthcare Changes Due to COVID-19 Since March 2020‡ | | 2361 (78.8) | 3272 (84.4) |
| Drinking Behavior Change Due to COVID-19‡ | Do Not Drink | 1185 (39.6) | 2051 (53.5) |
| | Drink More | 125 (4.2) | 163 (4.3) |
| | Drink Less | 189 (6.3) | 199 (5.2) |
| | Same | 1492 (49.9) | 1421 (37.1) |
| | BMI (kg/m²)†,‡ | 26.8±4.6 | 25.9±5.8 |
| Smoking Status‡ | Never | 1320 (44.0) | 2275 (58.7) |
| | Not currently, but in the past | 1568 (52.3) | 1505 (38.8) |
| | Yes | 110 (3.7) | 99 (2.6) |
| | Yes | 2,972 (99.2) | 3,837 (99.2) |
| | I don't know / not sure | 3 (0.1) | 2 (0.1) |
| Have Healthcare Coverage‡ | No | 21 (0.7) | 28 (0.7) |
| Has Primary Care Provider/Physician ‡ | | 2908 (97.0) | 3767 (97.4) |

*Comprised of a majority of Chinese Americans (38.1%) and Filipino (36.6%).
†Mean ± SD; other numbers are counts (and percentages).
‡For education attained, there are missing responses from 16 men and 27 women. For the other questions asked in the MEC baseline and COVID survey, the missing responses range from 35-51 for men and 57–106 for women.

**Table 2.** Distribution for postponing regular health care visits due to COVID-19 pandemic by sex (N=6974).

| | Male N (%) | Female N (%) |
|---|---|---|
| Postponed regular health care visits | 1625 (54.2) | 2369 (61.3) |
| Missing | 38 | 72 |
| | | |
| Primary care physician | 858 | 1188 |
| Oncologist | 54 | 105 |
| Cardiologist | 207 | 147 |
| Endocrinologist | 44 | 74 |
| Dentist | 954 | 1453 |
| Other | 768 | 1416 |

followed by Blood Pressure Medication ending in –sartan (29.3%; 26.0%), and Blood Pressure Medication ending in -pril (25.8%, 16.4%).

Overall, 54.2% of men and 61.2% of women participants reported that they had to postpone their regular healthcare visits due to COVID-19. Primary care physician and dentist visits were the most frequently postponed with 858 and 954 reports for men and 1188 and 1453 reports for women, respectively (*Table 2*).

Similarly, 5.7% of men and 11.0% of women reported having to postpone a cancer screening or procedure due to COVID-19 (*Table 3*). The screening procedure the most frequently missed was mammography (n=264), followed by skin examination (n=207) and colorectal cancer screening (n=114). In the past five years, 917 survey participants had been diagnosed with cancer (*Supplementary file 2*). Of note, 103 men (3.4%) and 135 women (3.4%) reported that they were currently receiving any of the following cancer treatments: chemotherapy (31 men; 30 women); immunotherapy: (22 and 19) and

**Table 3.** Distribution for postponing cancer screening test/procedure/treatment due to COVID-19 pandemic by sex (N=6974).

| | Male N (%) | Female N (%) |
|---|---|---|
| Postponed any cancer screening test/procedure | 169 (5.7) | 422 (11.0) |
| Missing | 48 | 87 |
| | | |
| Breast cancer screening (e.g. mammography, breast MRI) | - | 264 |
| Colorectal cancer screening (e.g. FOBT test, FIT test, colonoscopy, sigmoidoscopy) | 42 | 72 |
| Skin cancer screening (e.g. skin exam) | 83 | 124 |
| Prostate cancer screening (digital rectal examination, PSA) | 25 | - |
| Cervical cancer screening (e.g. PAP smear, pelvic exam)* | - | 18 |
| Imaging (e.g. CAT scan, DEXA, MRI)* | 12 | 24 |
| Other | 24 | 29 |
| | | |
| Currently on chemotherapy, immunotherapy, or radiation therapy for cancer | 103 (3.4) | 135 (3.4) |
| Chemotherapy | 31 | 30 |
| Immunotherapy | 22 | 19 |
| Radiation Therapy | 16 | 8 |
| | | |
| Postponed any cancer therapy session(s) | 7 (11.5) | 4 (6.9) |
| Missing | 2973 | 3882 |

*Categories created by sorting text responses in the 'Other Specify' answer choice.

**Table 4.** Odds Ratio* for change in lifestyle or daily activities since onset of COVID-19 pandemic by demographics and comorbidities.

| | Male (N=3000) | | | Female (N=3890) | | |
|---|---|---|---|---|---|---|
| | Yes (n) | Odds Ratio (95% CI) | p-Value | Yes (n) | Odds Ratio (95% CI) | p-Value |
| **Ethnicity** | | | | | | |
| White (N=3,031) | 1083 | 1.00 (Ref) | | 1,448 | 1.00 (Ref) | |
| Japanese American (N=2,228) | 833 | 1.02 (0.83,1.26) | 0.83 | 994 | 0.86 (0.69,1.07) | 0.17 |
| Latino (N=520) | 177 | 0.99 (0.70,1.40) | 0.96 | 233 | 0.94 (0.65,1.35) | 0.73 |
| Native Hawaiian (N=509) | 135 | 0.59 (0.42,0.84) | 0.003 | 235 | 0.44 (0.33,0.60) | <.0001 |
| African American (N=354) | 96 | 2.54 (1.30,4.99) | 0.01 | 211 | 0.96 (0.64,1.43) | 0.82 |
| Other (N=248) | 37 | 0.44 (0.25,0.78) | 0.01 | 151 | 0.62 (0.41,0.92) | 0.02 |
| Age (Years) (N=6,890) | 2361 | 0.98 (0.96,1.00) | 0.02 | 3272 | 0.97 (0.96,0.99) | 0.001 |
| Maximum Education Obtained (Years) (N=6,847) | 2347 | 1.12 (1.07,1.17) | <0.0001 | 3248 | 1.10 (1.06,1.15) | <0.0001 |
| **Comorbidities** | | | | | | |
| None (N=2,307)[†] | 694 | 1.00 (Ref) | | 1140 | 1.00 (Ref) | |
| Heart Disease (N=1,035) | 519 | 1.41 (1.11,1.80) | 0.01 | 356 | 1.19 (0.86,1.63) | 0.29 |
| Hypertension (N=3,271) | 1169 | 1.34 (1.11,1.62) | 0.002 | 1545 | 1.21 (1.00,1.47) | 0.05 |
| Diabetes (N=1,106) | 424 | 0.87 (0.68,1.10) | 0.25 | 462 | 0.90 (0.69,1.16) | 0.40 |
| Lung Disease, COPD, or Asthma (N=819) | 249 | 1.41 (1.01,1.96) | 0.04 | 466 | 1.65 (1.22,2.23) | 0.001 |
| Kidney Disease (N=340) | 138 | 1.37 (0.87,2.13) | 0.17 | 150 | 1.11 (0.70,1.76) | 0.65 |
| Diagnosed w/ Cancer in past 5 yrs. (N=921) | 383 | 1.09 (0.84,1.40) | 0.51 | 383 | 1.21 (0.89,1.64) | 0.22 |

*Adjusted for the other variables in the table. Male (Reference) vs. Female: OR = 1.62 (95% CI: 1.42,1.84; p<0.0001).
[†]Number of respondents who did not answer affirmatively for any listed co-morbidities.

radiation therapy (16 and 8). Among cancer patients still receiving treatment, 11.5% of men and 6.9% of women had to cancel at least one treatment session due to COVID-19 (*Table 3*).

Compared to White men, Native Hawaiian men were 41% (95% CI: 0.42, 0.84; p=0.003) and Other men were 56% (95% CI: 0.25, 0.78, p=0.004) less likely, and African American men were 154% (95% CI: 1.30, 4.99; p=0.007) more likely, to make any changes to their lifestyle or daily activities, such as covering a cough or sneeze, more frequent hand washing, and social distancing, since the COVID-19 pandemic (*Table 4*). Compared to White women, Native Hawaiian women were 56% (95% CI: 0.33, 0.60; p<.0001) and Other women were 38% (95% CI: 0.41, 0.92; p=0.02) less likely, to make any changes to their lifestyle or daily activities since the COVID-19 pandemic.

Each additional year of age was associated with less likelihood to make any changes to their lifestyle or daily activities by 2% per year of age for men (OR = 0.98, 95% CI: 0.96, 1.00; p=0.02) and by 3% per year of age for women (OR = 0.97, 95% CI: 0.96, 0.99; p=0.0005) (*Table 4*). Compared to men, women were 62% (95% CI: 1.42, 1.84; p<0.0001) more likely to make any changes to their lifestyle or daily activities. Each additional year of education was associated with more likelihood to make changes to their lifestyle or daily activities, by 12% per year of education in men (OR = 1.12, 95% CI: 1.07, 1.17; p<0.0001), and by 10% per year in women (OR = 1.10, 95% CI: 1.06, 1.15; p<0.0001).

Compared to men with no comorbidity, men with heart disease were 41% (95% CI: 1.11, 1.80; p=0.006), those with hypertension 34% (95% CI: 1.11, 1.62; p=0.002), and those with lung disease, COPD, or asthma 41% (95% CI: 1.01, 1.96; p=0.04), more likely to make changes to their lifestyle or daily activities (*Table 4*). Compared to women with no comorbidity, women with hypertension were 21% (95% CI: 1.00, 1.47; p=0.05), and those with lung disease, COPD, or asthma 65% (95% CI: 0.70, 1.76; p=0.001), more likely to make changes to their lifestyle or daily activities.

Similarly, compared to White men, Latino men were 26% (95% CI: 0.55, 0.99; p=0.04) and Native Hawaiian men 36% (95% CI: 0.47, 0.88; p=0.005) less likely to postpone regular health care visits due

**Table 5.** Odds ratio* for postponing regular health care visits due to COVID-19 pandemic by demographics and comorbidities.

| | Male (N=2998) | | | Female (N=3881) | | |
|---|---|---|---|---|---|---|
| | Yes (n) | Odds Ratio (95% CI) | p-Value | Yes (n) | Odds Ratio (95% CI) | p-Value |
| **Ethnicity** | | | | | | |
| White (N=3,026) | 770 | 1.00 (Ref) | | 1,042 | 1.00 (Ref) | |
| Japanese American (N=2,224) | 560 | 0.88 (0.74,1.04) | 0.13 | 715 | 0.93 (0.80,1.09) | 0.38 |
| Latino (N=519) | 114 | 0.74 (0.55,0.99) | 0.04 | 187 | 1.32 (0.99,1.76) | 0.06 |
| Native Hawaiian (N=508) | 88 | 0.64 (0.47,0.88) | 0.005 | 163 | 0.60 (0.47,0.78) | 0.0001 |
| African American (N=354) | 61 | 1.01 (0.67,1.52) | 0.98 | 155 | 1.01 (0.75,1.34) | 0.97 |
| Other (N=248) | 32 | 0.87 (0.51,1.49) | 0.62 | 107 | 0.77 (0.56,1.05) | 0.10 |
| Age (Years) (N=6,879) | 1,625 | 1.00 (0.99,1.02) | 0.85 | 2,369 | 0.98 (0.97,0.99) | 0.001 |
| Maximum Education Obtained (Years) (N=6,836) | 1,617 | 1.04 (1.00,1.08) | 0.04 | 2,354 | 1.03 (1.00,1.06) | 0.10 |
| **Comorbidities** | | | | | | |
| None (N=2,301)[†] | 445 | 1.00 (Ref) | | 800 | 1.00 (Ref) | |
| Heart Disease (N=1,032) | 382 | 1.37 (1.13,1.65) | 0.001 | 266 | 1.18 (0.94,1.49) | 0.16 |
| Hypertension (N=3,266) | 819 | 1.16 (1.00,1.35) | 0.06 | 1,141 | 1.16 (1.00,1.33) | 0.04 |
| Diabetes (N=1,106) | 314 | 1.15 (0.94,1.40) | 0.18 | 346 | 1.00 (0.82,1.22) | 1.00 |
| Lung Disease, COPD, or Asthma (N=821) | 179 | 1.25 (0.98,1.61) | 0.08 | 335 | 1.12 (0.92,1.36) | 0.26 |
| Kidney Disease (N=339) | 100 | 1.16 (0.83,1.62) | 0.40 | 123 | 1.49 (1.05,2.11) | 0.03 |
| Diagnosed w/ Cancer in past 5 yrs. (N=920) | 268 | 1.04 (0.85,1.27) | 0.73 | 305 | 1.51 (1.21,1.88) | 0.0003 |

*Adjusted for the other variables in the table. Male (Reference) vs. Female: OR = 1.43 (95% CI: 1.29,1.58; p<0.0001).

[†]Number of respondents who did not answer affirmatively for any listed co-morbidities.

to the COVID-19 pandemic (*Table 5*). Compared to White women, Native Hawaiian women were 40% (95% CI: 0.47, 0.78; p=0.0001) less likely to postpone regular healthcare visits due to the COVID-19 pandemic.

Each additional year of age in women was associated with a lower likelihood to postpone regular healthcare visits by 2% per year (OR = 0.98, 95% CI: 0.97, 0.99; p=0.0006) (*Table 5*). Each additional year of education in men was associated with a higher likelihood to postpone regular health care visits, by 45% per year (OR = 1.04, 95% CI: 1.00, 1.08; p=0.04). Compared to men, women were 43% (95% CI: 1.29, 1.58; p<0.0001) more likely to postpone regular health care visits.

Compared to men with no comorbidity, men with heart disease were 37% (95% CI: 1.13, 1.65; p=0.0011) more likely to postpone regular health care visits (*Table 5*). Compared to women with no comorbidity, women with hypertension were 16% (95% CI: 1.00, 1.33; p=0.04), those with kidney disease 49% (95% CI: 1.05, 2.11; p=0.03), and those diagnosed with cancer in the past 5 years 51% (95% CI: 1.21, 1.88; p=0.0003), more likely to postpone regular health care visits.

Compared to White men, Japanese American men were 72% (95% CI: 0.18, 0.45; p<0.0001) less likely to postpone any cancer screening test/procedure due to the COVID-19 pandemic (*Table 6*). Compared to White women, Japanese American women were 40% (95% CI: 0.46, 0.78; p=0.0002) less likely to postpone any cancer screening test/procedure due to the COVID-19 pandemic.

Women were 4% less likely for each additional year of age (95% CI: 0.94, 0.98; p=0.001) to postpone any cancer screening test/procedure (*Table 6*). Women were 10% more likely for each additional year of education (95% CI: 1.05, 1.16; p=0.0004) to postpone any cancer screening test/procedure. Compared to men, women were 137% (95% CI: 1.95, 2.88; p<0.0001) more likely to postpone any cancer screening test/procedure.

Compared to men with no comorbidity, men diagnosed with cancer in the past 5 years were 254% (95% CI: 2.52, 4.95; p<0.0001) more likely to postpone any cancer screening test/procedure (*Table 6*).

**Table 6.** Odds ratio* for postponing cancer screening test/procedure due to COVID-19 pandemic by demographics and comorbidities.

| | Male (N=2988) | | | Female (N=3866) | | |
|---|---|---|---|---|---|---|
| | Yes (n) | Odds Ratio (95% CI) | p-Value | Yes (n) | Odds Ratio (95% CI) | p-Value |
| Ethnicity | | | | | | |
| White (N=3,018) | 123 | 1.00 (Ref) | | 227 | 1.00 (Ref) | |
| Japanese American (N=2220) | 23 | 0.28 (0.18,0.45) | <0.0001 | 90 | 0.60 (0.46,0.78) | 0.0002 |
| Native Hawaiian (N=508) | 7 | 0.48 (0.21,1.05) | 0.07 | 32 | 0.73 (0.48,1.11) | 0.14 |
| African American (N=353) | 7 | 0.90 (0.40,2.03) | 0.81 | 27 | 1.01 (0.65,1.57) | 0.96 |
| Latino (N=509) | 8 | 0.48 (0.23,1.03) | 0.06 | 25 | 0.90 (0.57,1.44) | 0.66 |
| Other | 1 | 0.24 (0.03,1.80) | 0.17 | 21 | 0.86 (0.53,1.41) | 0.55 |
| Age (Years) (N=6854) | 169 | 0.96 (0.93,1.00) | 0.06 | 422 | 0.96 (0.94,0.98) | 0.001 |
| Maximum Education Obtained (Years) (N=6,811) | 166 | 1.03 (0.95,1.12) | 0.50 | 421 | 1.10 (1.05,1.16) | 0.0004 |
| Comorbidities | | | | | | |
| None (N=2295)[†] | 47 | 1.00 (Ref) | | 121 | 1.00 (Ref) | |
| Heart Disease (N=1028) | 29 | 0.82 (0.53,1.27) | 0.37 | 52 | 1.21 (0.87,1.69) | 0.27 |
| Hypertension (N=3255) | 75 | 1.07 (0.76,1.50) | 0.70 | 195 | 1.07 (0.86,1.34) | 0.56 |
| Diabetes (N=1099) | 22 | 0.88 (0.54,1.44) | 0.61 | 53 | 0.83 (0.59,1.16) | 0.27 |
| Lung Disease, COPD, or Asthma (N=815) | 17 | 0.84 (0.48,1.47) | 0.54 | 79 | 1.50 (1.14,1.98) | 0.004 |
| Kidney Disease (N=337) | 7 | 0.89 (0.40,2.00) | 0.78 | 18 | 0.94 (0.55,1.58) | 0.80 |
| Diagnosed w/ Cancer in past 5 yrs. (N=918) | 71 | 3.54 (2.52,4.95) | <0.0001 | 112 | 3.22 (2.50,4.14) | <0.0001 |

*Adjusted for the other variables in the table. Male (Reference) vs. Female: OR = 2.37 (95% CI: 1.95,2.88; p<0.0001).
[†]Number of respondents who did not answer affirmatively for any listed co-morbidities.

Compared to women with no comorbidity, women with lung disease, COPD, or asthma were 50% (95% CI: 1.14, 1.98; p=0.004) and those diagnosed with cancer in the past 5 years were 222% (95% CI: 2.50, 4.14; p<0.0001) more likely to postpone any cancer screening test/procedure.

## Discussion

The results of this study suggest unique relationships between participant characteristics, such as sex, age, years of education, race/ethnicity, and comorbidities with the likelihood of making changes to lifestyle or daily activities and of postponing regular health care visits and cancer screenings/treatments during the COVID-19 pandemic. Responses were compared between males and females and by race/ethnicity to describe the differences between these groups.

African American men (compared to White men), women (compared to men), men and women with more education, men and women with heart disease, hypertension, or lung disease, COPD, or asthma (compared to men or women with no comorbidity) were more likely to make changes to their lifestyle or daily activities during the COVID-19 pandemic. As expected, participants with one or more risk factors, such as cancer or lung disease, were more likely to make changes to their lifestyle during the pandemic as they were reported to be at increased risk of severe illness from COVID-19.

Men with more education, women (compared to men), men with heart disease, and women with hypertension, kidney disease, and diagnosed with cancer in the past 5 years were more likely to postpone regular health care visits. As expected, participants with comorbidity were more likely to postpone healthcare visits, as they were recommended to avoid public places to protect themselves from the virus.

Lastly, women with more education, women (compared to men), women and men diagnosed with cancer in the past 5 years, and women with lung disease, COPD, or asthma were more likely to postpone any cancer screening test/procedure.

Native Hawaiian men and women and those of Other race/ethnicity (compared to White men or women), and older men and women were less likely to make changes to their lifestyle or daily activities during the COVID-19 pandemic. Latino and Native Hawaiian men, Native Hawaiian women, and older women were less likely to postpone regular healthcare visits during the COVID-19 pandemic. Lastly, Japanese American men and women and older women were less likely to postpone any cancer screening test/procedure during the COVID-19 pandemic.

## Effect on healthcare visits

From May 2020 to September 2020, 54.2% of male and 61.3% of female MEC survey respondents reported postponing regular health care visits. These numbers are consistent with a report published by The Commonwealth Fund in May 2020 (*Mehrotra et al., 2020*). In the U.S., early in the pandemic from February 2020, the number of visits to ambulatory care practice declined by nearly 60% (*Mehrotra et al., 2020*). Declines in visits varied by geographic area and clinical specialty. By April 2020, a rebound of in-person visits had occurred and appears largest in the South Central (East and West) census division (Texas, Oklahoma, Arkansas, Louisiana, Mississippi, Alabama, Tennessee, and Kentucky) but the number of visits was still roughly one-third lower than before the pandemic. As in-person visits declined, telehealth visits increased rapidly before plateauing. By mid-May 2020, a rebound in visits had occurred across all specialties. But the relative decline in visits remained largest among surgical and procedural specialties and pediatrics. The greater effect on surgical and procedural specialties was observed in the MEC survey responses, as visits with eye specialists, dermatologists, cardiologists, and dentists, were more often reported to be postponed. In the Commonwealth Fund study, the relative decline was smaller in other specialties, such as adult primary care. The MEC study findings differ as visits for this specialty were the most frequently postponed in our data. In addition, researchers from Harvard Medical School examined the trends in outpatient care delivery and telemedicine in a cohort of 16,740,365 enrollees captured from the OptumLabs Data Warehouse. The weekly rate of telemedicine visits increased during the pandemic period starting from January 1, 2020, and peaking in the week of April 15, 2020, before declining by the week of June 10, 2020. By the last 4 weeks of the study period, May 20 through June 16, Hawai'i had a –73.2% change and California had a –31.0% change from baseline in weekly total visits per 1000 members. For visits delivered via telemedicine, Hawai'i had a 24.5% increase from baseline and California had a 29.5% increase (*Patel et al., 2021*). Although the COVID-19 pandemic led to widespread disruptions in medical care, it is possible that patients who avoided hospital visits had a positive outcome of lower risk of catching COVID-19. There may have been some benefit to the strategy of keeping patients out of the hospitals and physician offices. In the baseline MEC COVID Survey, only 49 men and 68 women, overall, reported an episode of COVID-19.

The likelihood of postponing healthcare visits being related to the presence of comorbidities was also investigated. In the MEC Survey, men with heart disease (by 37%), women with hypertension (by 16%), kidney disease (by 49%), and those diagnosed with cancer in the past 5 years (by 51%) were more likely to postpone regular health care visits due to the COVID-19 pandemic. These findings were consistent with a study that examined factors associated with postponed health checkups during the COVID-19 pandemic in Germany (*Hajek et al., 2021b*). Researchers found the probability of postponed health checkups was positively associated with the presence of chronic disease (OR: 1.68, 95% CI: 1.15, 2.47), higher concern for a COVID-19 infection (OR: 1.44, 95% CI: 1.16, 1.78), and higher presumed severity of COVID-19 (OR: 1.17, 95% CI: 1.01, 1.35).

## Effect on cancer screening

Not many MEC survey respondents reported postponing a cancer screening test or procedure due to COVID-19. Only 5.7% of men and 11.0% of women did so. These numbers were lower than expected but could be explained by the older age of our sample. The recommended cut-off age for breast and colorectal cancer screening is 75 years old. Half of the respondents were aged 75 and older. However, when focusing on participants 75 years and younger, the numbers were similar to the overall sample size: 6.1% vs 5.7% of men overall and 12.3% vs 11.0% of women overall. The screening procedure

the most frequently missed was mammography, followed by skin examination and colorectal cancer screening. Postponement of these cancer screening procedures is consistent with data from a published study looking at a sample of health claims clearinghouse records from 18 states containing 184 million claims from 30 million patients in 2019 and 94 million claims from 20 million patients for the first 6 months in 2020 (*Martin et al., 2022*). Mammograms and Pap smears were down nearly 80% in April 2020 compared to 2019. However, numbers for both services recovered throughout the summer and fall, with Pap smears and mammograms rebounding above 2019 levels in August 2020 and November 2020, respectively. Colonoscopies were down almost 90% in mid-April 2020, compared to 2019, and in December 2020 were still down about 15% compared to the previous year, representing a substantial but incomplete rebound in care delivered. The overall number of colonoscopies performed in 2020 declined by almost 25% from 2019.

The likelihood of postponing cancer screenings was also analyzed in relation to age. In the MEC study, women with each additional year of age were less likely by 4% per year of age (OR = 0.96, 95% CI: 1.00, 1.08; p=0.04) to postpone any cancer screening test/procedure. This finding is consistent with a study examining determinants of postponed cancer screening during the COVID-19 pandemic in Germany (*Hajek et al., 2021a*). Multiple logistic regressions were conducted with postponed cancer screenings since March 2020 due to the COVID-19 pandemic as the outcome measure (0=no, attended as planned, 1=postponed). Regressions revealed that the likelihood of postponing cancer screening was positively associated or more likely with higher concern for a COVID-19 infection (OR: 1.65, 95% CI: 1.16, 2.35), whereas it was negatively associated or less likely with older age (e.g. 65 years and over, OR: 0.38, 95% CI: 0.16, 0.89, compared to individuals 30–49 years).

Effective cancer screenings can lead to an early diagnosis of cancer and improve the 5-year relative survival rate. The CDC and the National Cancer Institute utilize a staging system called Summary Stage. Summary Stage groups invasive cancers as localized (the tumor is only in the organ it started in), regional (the tumor has spread to nearby organs, structures, or regional lymph nodes), distant (the tumor has spread to parts of the body far from where it started), and unknown (*Cent Dis Control Prev, 2023*). A study conducted in 2020 examined the 5-year relative survival rate for cancers diagnosed from 2009–2015 by race and stage of diagnosis. For breast cancer, the 5-year relative survival rate for all races in the localized, regional, and distant stages is 99%, 86%, and 27%, respectively (*Siegel et al., 2020*). For colorectal cancer, the 5-year relative survival rate for all races in the localized, regional, and distant stages is 90%, 71%, and 14%, respectively (*Siegel et al., 2020*). For both cancers, there is an obvious decline in the survival rate with progressive disease, which is concerning as a percentage of participants elected to postpone their cancer screenings, which could lead to a delay in diagnosis. However, it is important to note that our study was conducted during a four-month period at the height of the pandemic (May to September 2020) in the United States when vaccinations for COVID-19 hadn't been released. The Food and Drug Administration granted emergency use authorization to the Pfizer-BioNTech vaccine on December 10, 2020, a few months after the final baseline survey was collected (*FDA, 2021*). So our study does not address changes in health behaviors that may have occurred subsequent to the introduction of preventive vaccines.

The backlog of postponed or delayed cancer screenings will have a long-term clinical impact and potentially increase the cancer burden. A study from Canada examined the impact of episodic screening interruptions for breast and colorectal cancer (*Yong et al., 2021*). Using a simulation model, researchers projected a surge of cancer cases when screening resumes. For breast cancer screening, the simulation model suggests that a three-month interruption of breast cancer screening due to COVID-19 would result in 644,000 fewer screens being performed in Canada in 2020. A three-month interruption starting in 2020 could increase cases diagnosed at advanced stages (310 more) and breast cancer deaths (110 more) in Canada from 2020 to 2029. A six-month interruption starting in 2020 could lead to 670 extra advanced breast cancers and 250 additional breast cancer deaths in Canada from 2020 to 2029. For colorectal cancers, without service interruption, the simulation model estimated that 68,000 colonoscopies would have been performed in the six months since March 2020 in Canada. A six-month suspension of primary screening could increase cancer incidence by 2200 cases with 960 more cancer deaths over the lifetime. Longer interruptions, and reduced volumes of patients when screening resumes, would further increase excess cancer deaths. Since cancer diagnoses and deaths in the MEC are identified through linkages to cancer registries in HI and California, we will be able to examine whether a greater proportion of cancers were diagnosed at a late stage in

the aftermath of the pandemic and investigate any differences among ethnic/racial groups. Similarly, through the linkage of the cohort to Medicare, we will be able to identify MEC members who developed COVID-19 and investigate long-term complications and survival across age and race.

## Limitations

There are several limitations to our study. Although our study population is diverse, some subgroups of race/ethnicity (African Americans and Latinos) were under-represented compared to other subgroups. Additionally, the study sample is not fully representative of the entire MEC cohort, as the participants were more educated with higher percentages with at least some college education across all races/ethnicities. The survey respondents included a larger representation of Whites and Japanese Americans and a smaller representation of Latinos and African Americans compared to the MEC survivors in 2019. The effect of COVID-19 on cervical cancer screening was not investigated due to the older age of our participants. The data was self-reported so there is a chance of misclassification due to recall error. Lastly, the response rates were low across all races/ethnicities, and only 0.9% of male and 1.7% of female survey participants are living in a retirement or care home so these populations were most certainly underrepresented.

## Conclusion

The MEC COVID-19 survey demonstrated the possibility of using a mature cohort study of well-characterized individuals to characterize the effect of a public health emergency. The study revealed associations of factors like sex, race/ethnicity, age, education level, and comorbidities with cancer-related screening and healthcare among MEC participants during the COVID-19 pandemic in Hawai'i and Los Angeles. The MEC COVID-19 survey results were consistent with other studies regarding the postponement of healthcare visits, surgical procedures, and cancer screenings during the pandemic. In the wake of the pandemic, increased monitoring of patients in high-risk groups for cancer and other diseases is of the utmost importance as the chance of undiagnosed cases and poor prognosis due to delayed screening and/or treatment increases.

## Acknowledgements

This research was partially supported by the Omidyar 'Ohana Foundation and grant U01 CA164973 from the National Cancer Institute. Special thanks to Faye Nagamine, Yunoh Jung, and Anne Tome for assistance with data collection, data management, and statistical analysis.

## Additional information

### Funding

| Funder | Grant reference number | Author |
| --- | --- | --- |
| Hawaii Community Foundation | Omidyar 'Ohana Fund 20DA-101546 | Loic Le Marchand |
| National Cancer Institute | U01 CA164973 | Lynne R Wilkens<br>Christopher A Haiman<br>Loic Le Marchand |

The funders had no role in study design, data collection and interpretation, or the decision to submit the work for publication.

### Author contributions

Victoria P Mak, Data curation, Formal analysis, Visualization, Writing – original draft, Writing – review and editing; Kami White, Data curation, Formal analysis, Writing – review and editing; Lynne R Wilkens, Conceptualization, Supervision, Methodology, Writing – review and editing; Iona Cheng, Christopher A Haiman, Conceptualization, Methodology, Writing – review and editing; Loic Le Marchand, Conceptualization, Supervision, Funding acquisition, Methodology, Project administration, Writing – review and editing

Author ORCIDs
Victoria P Mak (iD) http://orcid.org/0000-0002-0908-5018
Christopher A Haiman (iD) http://orcid.org/0000-0002-0097-9971
Loic Le Marchand (iD) http://orcid.org/0000-0001-5013-980X

## Ethics

Human subjects: All participants provided informed consent before filling out the survey. The study was approved by the IRBs of the University of Hawaii (CHS 9575) and the University of Southern California (HS-17-00714).

## Decision letter and Author response

Decision letter https://doi.org/10.7554/eLife.86562.sa1
Author response https://doi.org/10.7554/eLife.86562.sa2

---

# Additional files

## Supplementary files

• Supplementary file 1. Comparison of Ethnicity and Educational Level between 2019 MEC Cohort Survivors and 2020 COVID Survey Participants.

• Supplementary file 2. Distribution of Baseline COVID Survey Participants by Health Status, Comorbidities, and Medication Use, Hawai'i and Los Angeles (N=6,974).

• Supplementary file 3. Distribution for Postponing Regular Health Care Visits Due to COVID-19 Pandemic by Sex (N=6,974).

• MDAR checklist

## Data availability

The Multiethnic Cohort welcomes applications from researchers to maximize the utility of the MEC data and/or specimens for prevention and etiological research on cancer and other chronic diseases. For access, a research application is required. To request access to the MEC resource for data analyses or ancillary studies, and to submit an application, please visit https://www.uhcancercenter.org/for-researchers/mec-data-sharing for further instructions. You will need to request an account in the system if you do not already have one. Proposals for access to MEC data or biospecimen are reviewed quarterly by the MEC Research Committee (MECRC) and must be submitted by the following deadlines: December 1, March 1, June 1, and September 1. All applications are reviewed by the MECRC following a standard procedure. The Statistical Analysis System (SAS), version 9.4 codes used for this study are available at https://github.com/MEC-COVID/eLife-Manuscript (copy archived at *Mak et al., 2023*). If you have questions please contact Gail Ichida at gichida@cc.hawaii.edu.

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
