## [Editor Report]

The authors used the Multiethnic Cohort (MEC) study to study how COVID-19 impacted access to cancer screenings and treatment. This study's important findings served to identify key factors associated with cancer-related screening and healthcare-seeking during the pandemic. This investigation provides solid evidence to inform future policies, particularly in older and vulnerable populations.

---

## [Decision Letter]

**Decision letter after peer review:**

Thank you for submitting your article "The Impact of COVID-19 on Cancer Screening and Treatment in Older Adults: The Multiethnic Cohort Study" for consideration by *eLife*. Your article has been reviewed by one peer reviewer, and I oversaw the evaluation in my dual role of Reviewing Editor and Senior Editor.

Essential revisions:

I compiled below the comments from the reviewer in their critique and in interactions post-review. Please submit a revised version that addresses these concerns directly. Although we expect that you will address these comments in your response letter, we also need to see the corresponding revision clearly marked in the text of the manuscript. Some of the comments may seem to be simple queries or challenges that do not prompt revisions to the text. Please keep in mind, however, that readers may have the same perspective as the reviewer. Therefore, it is essential that you amend or expand the text to clarify the narrative accordingly.

*Reviewer #1 (Recommendations for the authors):*

This study takes advantage of a cohort (the multi-ethnic cohort) and conducted a survey to understand various measures that might be associated with delays to cancer screening and healthcare-seeking behaviors. Overall, this is a very good study, and the authors acknowledge the limitations of their work. One thing that might be added to the discussion is the temporal changes in healthcare-seeking behaviors, including early diagnosis of cancer or other screenings. As we know, the changes in infection rates over time and eventual access to preventive vaccines made a huge difference in the public's confidence to seek and attend healthcare; I think the author should comment on this since this work was conducted within six months during the height of the pandemic, and before vaccination.

---

## [Author Response]

Essential revisions:Reviewer #1 (Recommendations for the authors):One thing that might be added to the discussion is the temporal changes in healthcare-seeking behaviors, including early diagnosis of cancer or other screenings. As we know, the changes in infection rates over time and eventual access to preventive vaccines made a huge difference in the public's confidence to seek and attend healthcare; I think the author should comment on this since this work was conducted within six months during the height of the pandemic, and before vaccination.

Commentary on the temporal changes in healthcare-seeking behaviors and access to preventive vaccines during the pandemic has been added to the discussion. We have clarified this as follows:

“Effective cancer screenings can lead to an early diagnosis of cancer and improve the 5-year relative survival rate. The CDC and the National Cancer Institute utilize a staging system called Summary Stage. Summary Stage groups invasive cancers as localized (the tumor is only in the organ it started in), regional (the tumor has spread to nearby organs, structures, or regional lymph nodes), distant (the tumor has spread to parts of the body far from where it started), and unknown.^18^ A study conducted in 2020 examined the 5-year relative survival rate for cancers diagnosed from 2009 to 2015 by race and stage of diagnosis. For breast cancer, the 5-year relative survival rate for all races in the localized, regional, and distant stages is 99%, 86%, and 27%, respectively.^9^ For colorectal cancer, the 5-year relative survival rate for all races in the localized, regional, and distant stages is 90%, 71%, and 14%, respectively.^9^ For both cancers, there is an obvious decline in the survival rate with progressive disease, which is concerning as a percentage of participants elected to postpone their cancer screenings, which could lead to a delay in diagnosis. However, it is important to note that our study was conducted during a four-month period at the height of the pandemic (May to September 2020) in the United States when vaccinations for COVID-19 hadn’t been released. The Food and Drug Administration granted emergency use authorization to the Pfizer-BioNTech vaccine on December 10, 2020, a few months after the final baseline survey was collected.^19^ So our study does not address changes in health behaviors that may have occurred subsequent to the introduction of preventive vaccines.”